On reconstructing Giraffa sivalensis, an extinct giraffid from the Siwalik Hills, India

van Sittert Sybrand J. 1 sybrandvs@gmail.com
Mitchell Graham 1 2
1 Centre for Veterinary Wildlife Studies, Department of Production Animal Studies, Faculty of Veterinary Science, University of Pretoria , South Africa
2 Department of Zoology and Physiology, University of Wyoming , Laramie Wyoming , USA
Hutchinson John
Electronic publication date: 2015 Aug 4
Publication date: 2015
Volume: 3
Electronic Location ID: e1135
Received 2015 Mar 13; Accepted 2015 Jul 7
Copyright: © 2015 van Sittert and Mitchell
Copyright year: 2015
Copyright holder: van Sittert and Mitchell
License: This is an open access article distributed under the terms of the Creative Commons Attribution License, which permits unrestricted use, distribution, reproduction and adaptation in any medium and for any purpose provided that it is properly attributed. For attribution, the original author(s), title, publication source (PeerJ) and either DOI or URL of the article must be cited.
License URL: https://creativecommons.org/licenses/by/4.0/

Keywords: Giraffa, Allometry, Neck length, Giraffidae, Plio-Pleistocene, Scaling, Body mass, Siwalik, Okapia

Funding: University of Pretoria Bursary Maberly Memorial Scholarship Don Craib Trust University of Wyoming SJ Van Sittert received a University of Pretoria Bursary as well as the Maberly Memorial Scholarship. Additional funds were provided by the Don Craib Trust (John Skinner, deceased), from a personal research grant (John Skinner), and the University of Wyoming (Graham Mitchell). The funders had no role in study design, data collection and analysis, decision to publish, or preparation of the manuscript.

==============================
Giraffa sivalensis occurred during the Plio-Pleistocene period and probably represents the terminal species of the genus in Southern Asia. The holotype is an almost perfectly preserved cervical vertebra of disputed anatomical location. Although there is also uncertainty regarding this animal’s size, other specimens that have been assigned to this species include fragments of two humeri, a radius, metacarpi and teeth. Here we estimate neck length, leg length and body mass using interspecific and, unusually, ontogenetic allometry of extant giraffe skeletal parameters. The appropriateness of each equation to estimate body mass was evaluated by calculating the prediction error incurred in both extant giraffes (G. camelopardalis) and okapis (Okapia johnstoni). It followed that the equations with the lowest prediction error in both species were considered robust enough to use in G. sivalensis. The size of G. sivalensis, based on the holotype, is proposed as 400 kg (range 228 kg–575 kg), with a neck length of approximately 147 cm and a height of 390 cm. The molar lengths of tooth specimens considered agree with this size estimate. The humerus was the most appropriate long bone to establish body mass, which estimates a heavier animal of ca 790 kg. The discrepancy with the vertebral body weight estimate might indicate sexual dimorphism. Radial and metacarpal specimens estimate G. sivalensis to be as heavy as extant giraffes. This may indicate that the radius and metacarpus are unsuitable for body mass predictions in Giraffa spp. Alternatively, certain long bones may have belonged to another long legged giraffid that occurred during the same period and locality as G. sivalensis. We have concluded that if sexual dimorphism was present then males would have been about twice the size of females. If sexual dimorphism was not present and all bones were correctly attributed to this species, then G. sivalensis had a slender neck with a relatively stocky body.

Introduction

Giraffa sivalensis (Falconer & Cautley, 1843) was the first extinct Giraffa species to be discovered, yet neither a complete skull nor specimens related to the holotype vertebra have been found. Notwithstanding this limitation, many fossil specimens have been assigned as belonging to this species, without adequate consideration of its size or without explicitly citing the stratigraphic horizon of discovery (Table S1). In addition, many of the discovered specimens have only been described in the Fauna Antiqua Sivalensis, which is a collection of Falconer’s publications and unpublished notes (Murchison, 1868b). Although all the plates (notably plate E) within the Fauna Antiqua Sivalensis are well described (Murchison, 1868a), many of them have never been published.

History of G. sivalensis discovery

Cautley (1838) briefly described the discovery of a remarkable vertebra in the Siwalik Hills in India. He believed the specimen to be very similar to that of extant giraffes—a significant statement, because up until that time no other fossil Giraffa species were known. Falconer & Cautley (1843) subsequently named the species Camelopardalis sivalensis and assigned the fossil, which was to become the holotype (Badam, 1979), as a third cervical vertebra. However, Lydekker (1885) disputed this and proposed that the holotype was in actual fact a fifth cervical vertebra of a ‘very small individual’. Since Cautley’s discovery, other Giraffa-like fossils have also been found in Asia, Europe and Africa, subsequently leading to proposals for species such as G. priscilla, G. jumae, G. stillei, G. gracilis, G. pygmaea and G. punjabiensis. The references to these fossil specimens are extensive, incomplete and confusing as can be seen by the references to G. sivalensis alone in Table S1.

Geographic and stratigraphic distribution of fossils

Matthew (1929) placed the upper Siwalik deposits, where G. sivalensis fossils and nearly all Siwalik fauna discovered by early writers such a Falconer have been found (Lydekker, 1876), as part of the Pinjor zone (Gaur, Vasishat & Chopra, 1985; Akhtar et al., 1991; Nanda, 2002; Nanda, 2008; Bhatti, 2004). The Pinjor zone dates to roughly 2.58–0.6 million years ago, placing the fauna discovered in this site as originating during the late Pliocene/early Pleistocene (Nanda, 2008). The site of discovery of the holotype for G. sivalensis was presented by Falconer & Cautley (1843) only as ‘the Sewalik range to the west of the river Jumna’ (currently the Yamuna river). Although Spamer, Daeschler & Vostreys-Shapiro (1995) described the locality as ‘Siwalik Hills, near Hardwar, Uttar Pradesh’, this is unlikely as Hardwar is east of the Yamuna. We therefore believe the locality was probably in the vicinity of the current Shivalik fossil park, Saketi, Himachal Pradesh, India (Fig. 1).

Figure 1 A map indicating the probable vicinity of G. sivalensis fossil discoveries.

The marker indicates the location of the Shivalik Fossil Park in the Siwalik Hills, a subHimalayan mountain range. This is most probably the area ‘west to the river Jumna’ (currently Yamuna River) to which Falconer & Cautley (1843) referred. Map data: AutoNavi, Google.

Size estimates and controversy

Size estimates of G. sivalensis have been inadequate or contradictory. For example, it has been proposed that G. sivalensis was about ‘one third shorter’ with a neck about ‘one tenth more slender’ than extant giraffes (Falconer & Cautley, 1843), and that the holotype belonged to a very small individual (Lydekker, 1885), that it had the same sized cranium as extant giraffes but with a shorter neck (Lydekker, 1876), that it was a large species but smaller than extant giraffes (Bhatti, 2004, p. 155), that it was of comparable size to modern giraffes (Bhatti, 2004, p. 255), that it was larger than extant giraffes (Mitchell & Skinner, 2003) and that certain proportions of the species’ neck were larger than extant giraffes (Lydekker, 1876, p. 105). Additional fossil specimens originally thought to belong to a separate species, G. affinis (Falconer & Cautley, 1843), were subsequently shown to belong to G. sivalensis and are currently believed to indicate a larger individual of the species (Lydekker, 1876, p. 105; Bhatti, 2004, p. 140). Table 1 summarises previous size estimates for G. sivalensis.

Table 1 Previous size estimates of G. sivalensis.

Size estimate	Author	Relevant specimens/comments	
‘One third shorter’ with a neck ‘one tenth more slender’ as extant giraffes.	Falconer & Cautley, 1843;
Lydekker, 1876, p. 105	Holotype vertebra, OR39747.	
Large species but smaller than extant giraffes.	Bhatti, 2004, p. 155	No specimen referred to.	
Of comparable size to modern giraffes.	Bhatti, 2004, p. 225	No specimen referred to.	
Similar head size to extant giraffes but with a shorter neck.	Lydekker, 1876, p. 105	OR39747. Lydekker noted that the areas of the zygoapohyses are ‘considerably larger’ than in those of extant giraffes, making the neck ‘at least equally strong’ as that of extant giraffes. The larger cranial and caudal articular surfaces were also noted by Falconer & Cautley (1843).	
Similar in size to extant giraffes.	Lydekker, 1883	Cervical vertebra similar in size as that of G. camelopardalis. Referring to an imperfect ‘first’ cervical vertebra, later catalogued as a ‘third’ cervical, BM39746 (Lydekker, 1885).	
Slightly larger than extant giraffes.	Murchison, 1868b, p. 207	Right humerus. Museum of the Asiatic Society of Bengal no 43, Natural History Museum no 39749. Exact form to that of extant giraffes, but a little larger (Falconer, 1868). Lydekker (1885) however mentioned that this fossil bone originated from a ‘small individual’.	
Similar in size to extant giraffes.	Murchison, 1868b, p. 206	Left radius. Asiatic Museum of Bengal no 690. Nearly equal in dimensions to existing giraffes.	
Similar in size to extant giraffes.	Murchison, 1868b, p. 207	Left metacarpus. Asiatic Museum of Bengal no 52. Of the size of existing giraffe.	
Similar in size to extant giraffes.	Lydekker, 1885	Phalangeals, no 17131a. Almost indistinguishable from the corresponding bones of extant giraffes.	
Similar in size to extant female giraffes.	Falconer & Cautley, 1843	Fragments from upper and lower jaws. Falconer originally ascribed these specimens to G. affinis. Lydekker (1876) however refuted this species and proposed that it in actual fact G. sivalensis.	
Larger than extant giraffes with smaller teeth than extant giraffes.	Mitchell & Skinner, 2003	Review of literature.	

In this paper we outline and clarify the relevant information about G. sivalensis and its remains. In addition, we have made new estimates of its size and shape.

Materials and Methods

Studied material and dimensions measured

All postcranial specimens assigned to G. sivalensis that were available at the Natural History Museum in London were studied. From these specimens, body and neck size estimates were calculated using giraffe ontogenetic or available interspecific allometric equations. The only vertebra measured was the holotype (OR39747, Fig. 2), a cervical which had been extensively described by Falconer & Cautley (1843). A caudal fragment of a ‘fourth’ cervical (OR39748; Lydekker, 1885), also described as a second cervical by Falconer (1845), as well as a caudal part of a ‘third’ cervical (OR39746; Lydekker, 1885) were missing from the Siwalik collection at the Natural History Museum. Dimensions were measured with a vernier calliper and included: vertebral body length, cranial vertebral body height, cranial vertebral body width, caudal vertebral body height, caudal vertebral body width and spinous process length (Fig. 2).

Figure 2 Giraffa sivalensis holotype, specimen OR39747.

Presented, from left to right, in left lateral (A), right lateral (B), cranial (C) and caudal (D) views. On left lateral view the line indicates the landmarks for the vertebral body length (L) measurement. On cranial and caudal views the vertical lines indicate the height (dorsoventral, DV) while the horizontal lines indicate the width (transverse, T) measurements.

Additional postcranial specimens assigned to G. sivalensis held at the Natural History Museum include fragments of two humeri (OR39749 and OR17136; Figs. 3 and 4 respectively), a fragment of a radius/ulna (OR17130) and various fragments of metacarpi and phalanges. All metacarpal specimens except OR39750 were avoided due to the unclear numbering of specimens and deformation of the fossils. Measurements of the long bones included length, midshaft circumference and midshaft diameter in craniocaudal and transverse planes. The length and circumference measurements were done with a measuring tape, while the cross sectional diameters were done with a vernier calliper.

Figure 3 Specimen OR39749.

This image represents different views of a right humerus that has been assigned to G. sivalensis. The different views are not to scale; where only distal parts of the bone are shown, these have been enlarged relative to images of the specimen in toto. The scale bar indicates 50 mm and pertains to the lateral view only.

Figure 4 Specimen OR17136.

This represents different views of the proximal part of a left humerus that has been assigned to G. sivalensis. The scale bar indicates 50 mm and pertains to the lateral view only as the different views are not drawn to scale.

Because there is no complete G. sivalensis skeleton its shape needs to be inferred as analogous to the only other extant Giraffa: G. camelopardalis. One of the methods of inferring body size from a model animal or animals requires that regression equations in the form y = mxb (Huxley, 1932) be constructed. These regression equations can be based on data from different species (interspecific allometry), within the growth phase of a single animal (ontogenetic allometry) or amongst adult animals of different size but within the same species (static allometry). We applied ontogenetic as well as interspecific allometric equations to predict body mass in this case.

Ontogenetic data were obtained from previous studies by the authors (Mitchell, Van Sittert & Skinner, 2009; Van Sittert, Skinner & Mitchell, 2010; Van Sittert, Skinner & Mitchell, 2015). These data were used to construct allometric equations to describe body mass or body dimensions. The dimensions used from ontogenetic vertebral data are summarised in Table 2. Interspecific regression equations were sourced from previously published work (Anderson, Hall-Martin & Russell, 1985; Roth, 1990; Scott, 1990; Campione & Evans, 2012). The dimensions measured for the long bone ontogenetic data are summarised in Table 3.

Table 2 Dimensions for the G. sivalensis holotype; a well preserved third cervical vertebra (OR39747).

Falconer & Cautley’s (1843) findings are also presented. All values in mm. Nomenclature is based on the Nomina Anatomica Veterinaria (International Committee on Veterinary Gross Anatomical Nomenclature, 2012).

Dimension and description	Falconer & Cautley (1843)’s terminology	Present study’s measurement (±95% confidence interval for three measurements) (mm)	Falconer & Cautley (1843) measurement (mm)	
Vertebral body length: Longitudinal axis of the vertebral body (Corpus vertebrae), from the most cranial curvature of the cranial extremity (Extremitas cranialis (Caput vertebrae)) to the most caudal part of the caudal extremity (Extremitas caudalis (Fossa vertebrae))	Length of the body of the vertebrae between articulating heads.	200.2 ± 0.7	198.1	
Cranial vertebral body height: Greatest dorsoventral height of cranial extremity.	Vertical height articulating head?	42.9 ± 1.4	25.4	
Antero-posterior diameter articulating head?	48.3	
Cranial vertebral body width: Greatest transverse width of cranial extremity.	Greatest diameter at articulating head	36.2 ± 2.8	35.6	
Caudal Vertebral body height: Greatest dorsoventral height of caudal extremity.	Vertical diameter, articular cup, posterior end	53.1 ± 0.3	50.8	
Caudal vertebral body width: Greatest transverse width of caudal extremity.	Transverse diameter, articular cup, posterior end	53.4 ± 0.3	50.8	
Spinous process length: From roof of the vertebral foramen to the highest point of the spinous process, perpendicular to the long axis of the vertebral body.		21.8 ± 2.6		

Table 3 Dimensions for long bone specimens marked as belonging to G. sivalensis.

All values in mm. OR39749 is marked as a juvenile.

Specimen no	HL	HCirc	HCr	HTr	RL	RCirc	RCr	RTr	McL	MCirc	McCr	McTr	
OR39750a									389	186	53	60	
OR17130b					220	217	53	71					
OR39749c	453	212	66	66									
OR17136a	279	216	76	57									
Notes.

AbbreviationsH Humerus

R Radius

Mc Metacarpus

L Length

Circ Midshaft circumference

Cr midshaft craniocaudal diameter

Tr midshaft transverse diameter

a Distal proportion lacking.

b Only diaphysis.

c Proximal metaphysis missing.

There are inherent problems associated with using dental measurements as body size predictors, especially when only a single tooth is used (Damuth, 1990; Fortelius, 1990; Janis, 1990). Nevertheless, we have estimated size from teeth originally measured by Falconer & Cautley (1843), even though these teeth were initially assigned to a new species G. affinis, a species that was eventually abandoned (Lydekker, 1883). Uncertainty regarding these teeth specimens persisted until recent times (Spamer, Daeschler & Vostreys-Shapiro, 1995). Teeth specimens described by authors other than Falconer and Cautley which are noted in Table S1 were not evaluated further as there was either uncertainty regarding the authors’ species association (Lydekker, 1876), or the teeth specimens were not necessarily collected in the vicinity or stratigraphical layer of fossils described by Falconer and Cautley (Lydekker, 1878), or because certain specimens were deciduous. Table 4 presents dental specimens as well as dimensions as measured by Falconer & Cautley (1843). Body masses were estimated from regression equations established by Damuth (1990).

Table 4 Summary of fossil teeth assigned to G. affinis by Falconer & Cautley (1843), and subsequently assigned to G. sivalensis.

All regression equations were obtained from Damuth (1990).

Fossil specimens	Museum no	References to specimen	Dimensions	Relevant regression equation (reference)	Body mass prediction	
Fragment of left maxilla including two rear molars. The ‘back part of the maxillary, beyond the teeth, is attached’.	39756 a (Lydekker, 1885)	Figured in Plate 2 Figs. 3A and 3B of Falconer & Cautley (1843).	Joint length of two back molars, maxilla =2.5 in = 63.5 mm			
Greatest width of last molar = 1.4 in = 35.56 mm	38.02 × TUMW∧2.77 (all ungulates)	752 kg	
	32.36 × TUMW∧2.87 (all selenodonts)	945 kg	
	17.78 × TUMW∧2.97 (selenodont browsers)	718 kg	
Greatest width of penultimate molar =1.45 in = 36.83 mm	32.36 × SUMW∧2.78 (all ungulates)	731 kg	
	22.91 × SUMW∧2.96 (all selenodonts)	991 kg	
	12.02 × SUMW∧3.08 (selondont browsers)	801 kg	
Average of width measurements (SD)		823 (117) kg	
Rear molar of right maxilla	39756 (Lydekker, 1885)	Figured in Plate 2 Fig. 4 of Falconer & Cautley (1843).	Length = 1.2 in = 30.48 mm * it is not sure whether this is the greatest dimensions or occlusal surface.	19.50 × TUML∧2.81 (all ungulates)	288 kg	
	8.71 × TUML∧3.12 (all selenodonts)	372 kg	
	6.31 × TUML∧3.29 (selenodont browsers)	481 kg	
Average of length measurements (SD)		380 (97) kg	
Width = 1.4 in = 35.56 mm * it is not sure whether this is the greatest dimensions or occlusal surface.	38.02 × TUMW∧2.77 (all ungulates)	752 kg	
	32.36 × TUMW∧2.87 (all selenodonts)	915 kg	
	17.78 × TUMW∧2.97 (selenodont browsers)	718 kg	
Average of width measurements (SD)		795 (105) kg	
Fragment of left mandible containing the third molar	39755 (Lydekker, 1885)	Figured in plate 2 Figs. 5A and 5B of Falconer & Cautley (1843).	Length = 1.7 in = 43.18 mm	6.31 × TLML∧2.99 (all ungulates)	489 kg	
	3.24 × TLML∧3.19 (all selenodonts)	533 kg	
	2.24 × TLML∧3.35 (selenodont browsers)	673 kg	
Average of length measurements (SD)		565 (96) kg	
Greatest width =1.0 in = 25.4 mm	109.64 × TLMW∧2.73 (all ungulates)	750 kg	
	77.62 × TLMW∧2.93 (all selenodonts)	1,014 kg	
	64.56 × TLMW∧2.88 (selenodont browsers)	718 kg	
Average of width measurements (SD)		827 (162) kg	
Third premolar of the left mandible, detached.	39757 (Lydekker, 1885)	Figured in Plate 2 Fig. 6 of Falconer & Cautley (1843).	Length = 1.0 in = 25.4 mm	79.43 × TLPL∧2.76 (all ungulates)	599 kg	
	61.66 × TLPL∧2.92 (all selenodonts)	780 kg	
	20.42 × TLPL∧3.19 (selenodont browsers)	618 kg	
Average of length measurements (SD)		666 (99) kg	
Width = 0.9 in = 22.86 mm	524.81 × TLPW∧2.45 (all ungulates)	1,121 kg	
	524.81 × TLPW∧2.53 (all selenodonts)	1,440 kg	
	398.11 × TLPW∧2.49 (selenodont browsers)	964 kg	
Average of width measurements (SD)		1,175 (243) kg	
Second premolar of right maxilla		Figured in Plate 2 Fig. 7 of Falconer & Cautley (1843).	Length = 1.0 in = 25.4 mm	169.82 × SUPL∧2.51 (all ungulates)	570 kg	
		141.25 × SUPL∧2.65 (all selenodonts)	746 kg	
		20.41 × SUPL∧3.26 (selenodont browsers)	776 kg	
	Average of length measurements (SD)		697 (111) kg	
	Width =1.12 in = 28.45 mm	380.19 × SUPW∧2.3 (all ungulates)	840 kg	
		416.87 × SUPW∧2.31 (all selenodonts)	953 kg	
		208.93 × SUPW∧2.44 (selenodont browsers)	738 kg	
	Average of width measurements (SD)		843 (108) kg	
Notes.

AbbreviationsTUML Third Upper Molar Length

TUMW Third Upper Molar Width

SUMW Second Upper Molar Width

TLML Third Lower Molar Length

TLMW Third Lower Molar Width

TLPL Third Lower Premolar Length

TLPW Third Lower Premolar Width

SUPL Second Upper Premolar Length

SUPW Second Upper Premolar Width

SD Sample Standard Deviation

Statistical analyses

Allometric equations were generated from bivariate data through ordinary least squares regression. To facilitate this, measurements were logarithmically transformed to base e prior to analyses. According to Warton et al. (2006), ordinary least squares regression is appropriate when one wishes to predict y from x, even when x contains measurement error, as long as the results are interpreted in the context of ‘predicting y from x measured with error’. It is worth noting that there is controversy regarding the practice of logarithmically transforming data in scaling studies (Packard, Boardman & Birchard, 2009; Packard, Boardman & Birchard, 2010; Cawley & Janacek, 2010; Packard, 2013). The main argument is whether error becomes larger as body mass increases (multiplicative error), in which case logarithmic transformation is appropriate, or whether there is no correlation between error and body mass, in which case logarithmic transformation is not appropriate (Glazier, 2013). The debate is ongoing and will not be reviewed here. In this study we selected the method of log-transformation of data as it enables more convenient comparison among similar datasets.

Because body dimensions (especially body masses) can be predicted by different equations and by different fossil specimens, the predictions need to be validated. If regression equations had reasonable power in estimating body mass in both extant giraffids (G. camelopardalis and O. johnstoni), then they were regarded as robust enough to extrapolate to G. sivalensis as well. Therefore, dimensions of 10 okapi skeletons were recorded in addition to data obtained from G. camelopardalis. The okapi skeletons were housed in various museums and were recorded as the opportunities presented themselves (Table 5). Adult okapi specimens were assumed to have weighed 250 kg, with a range of 200 kg–300 kg (Lindsey & Bennett, 1999; Stuart & Stuart, 2006). The mature okapi specimens were identified through additional data associated with each museum specimen as well as by the degree of fusion of the epiphyses. The robustness of giraffe ontogenetic as well as interspecific equations to predict body mass in both adult giraffes and adult okapis correctly were assessed through the percent prediction error, calculated according to Smith (1984) and Van Valkenburgh (1990): Observed value−Predicted value/Predicted value×100.

Table 5 The studied okapi specimens and their dimensions used in determining the appropriateness of allometric equations in determining body size and shape estimates in G. sivalensis.

Specimen no	Museum	OTVL	OVNL	OVNL-1	C3VBL	OFL	N:FL	PVNL	Predicted neck length regression equation	% PE	
az2348	DMNH	1,259	557	522	85	932	0.60	586	PVNL = 10.65 ∗ C3VBL∧0.902	0.05	
az2440	DMNH	1,392	567	531	83			574	0.01	
1973-178	MNHN	722	273	260	42	752	0.36	310	0.14	
1961-131	MNHN	400	149	137	22.1	553	0.27	174	0.17	
1984-56	MNHN		459	428	73.5			514	0.12	
1996-102	MNHN	1,529	632	600	96.9	1,018	0.62	660	0.04	
27194	SM	1,442	621	589	106	1,018	0.61	715	0.15	
73224	SM	1,521	647	613	107	993	0.65	722	0.12	
56346	SM	1,458	630	599	102	998	0.63	691	0.10	
92290	SM			142	22	534				
Notes.

AbbreviationsDMNH Ditsong National Museum of Natural History (Formerly Transvaal Museum), Pretoria, South Africa

MNHN Museum National d’Histoire Naturelle, Paris, France

SM Senckenberg Naturmuseum, Frankfurt, Germany

OTVL Observed Total Vertebral Length

ONL Observed Neck Length

ONL-1 Observed Neck Length Minus C1

OTL Observed Trunk Length

OFL Observed Front Limb Long Bone Lengths

OHL Observed Hind Limb Long Bone Lengths

N:FL Neck Length to Foreleg Length ratio

PNL Predicted Neck Length

% PE Percent Prediction Error for vertebral length based on giraffe ontogenetic allometry

Assumptions made

One of the major assumptions of this study is that G. sivalensis dimensions can be modelled from G. camelopardalis ontogeny. Although it is unusual to model an animal from the ontogeny of a different species it is not unique (for example, Roth, 1990). In assigning the holotype to a specific vertebra, we also assumed that there would be broad similarity in shape between the cervical vertebrae of G. sivalensis and G. camelopardalis. Falconer & Cautley (1843) illustrated this assumption to be the case for many but not all features of the holotype vertebra.

Another assumption was that the specimens used came from the same Giraffa species. We tried to use only those specimens that were clearly attributable to the Plio-Pleistocene and to the vicinity of the holotype discovery (Fig. 1), to limit possible confusion with other Giraffa species like G. punjabiensis. However, in some instances these criteria were not clear due to the lack of other samples or information, as in the discussion of vertebrae OR39746 and OR39748. Lastly, in terms of estimating body proportions in adult animals based on vertebral length, similarity in shape to G. camelopardalis was assumed.

Results

Dimensions measured

The OR39747 and long bone dimensions measured are summarised in Tables 2 and 3 respectively, and where applicable the dimensions contain the equivalent measured values according to Falconer & Cautley (1843). Except for the cranial vertebral body height, our measurements on OR39747 are within 1%–5% of that reported by Falconer & Cautley (1843). Dimensions measured from okapi skeletons are presented in Table 5. Table 5 also contains predictions and prediction errors for okapi vertebral neck length based on G. camelopardalis ontogenetic data.

Predictions based on vertebra OR39747

Based on G. camelopardalis ontogenetic data, the average of dorsal and ventral neck length including soft tissue in G. sivalensis was 1,467 mm (y = 1.55x0.859), the vertebral neck length excluding soft tissue was 1,270 mm (y = 10.66x0.902) and the foreleg (hoof to withers) height in the living G. sivalensis adult was 2,540 mm (y = 7.61x0.663, Table 6). This would mean that the reaching height of G. sivalensis was around 3.9 m.

Table 6 Power functions, their origin and predicted values for linear dimensions of G. sivalensis.

Dimension predicted for G. sivalensis (dependent (y) variable)	Prediction based on (independent (x) variable)	Equation generated from	Equation, slope confidence interval, R2	Prediction	
Vertebral neck length (C1–C7)	OR39747 (C3) vertebral body length	G. camelopardalis ontogenetic data	y = 10.66x0.902 CI = 0.874–0.930 R2 = 0.99	1,270 mm	
Vertebral neck length (C2–C7)	OR39747 (C3) vertebral body length	G. camelopardalis ontogenetic data	y = 9.708x0.908 CI = 0.881–0.936 R2 = 0.99	1,195 mm	
Vertebral neck length (C2–C7)	OR39747 (C3) vertebral body length	Various ungulates, data from Badlangana, Adams & Manger (2009)	y = 5.023x1.025 CI = 0.977–1.614 R2 = 0.99	1,148 mm	
Dorsal neck length (occipital crest to withers)	OR39747 (C3) vertebral body length	G. camelopardalis ontogenetic data	y = 1.694x0.822 CI = 0.716–0.928 R2 = 0.87	1,321 mm	
Ventral neck length (angle of jaw to acromion)	OR39747 (C3) vertebral body length	G. camelopardalis ontogenetic data	y = 1.442x0.890 CI = 0.765–1.014 R2 = 0.85	1,608 mm	
Average neck length (of dorsal and ventral neck length)	OR39747 (C3) vertebral body length	G. camelopardalis ontogenetic data	y = 1.55x0.859 CI = 0.767–0.951 R2 = 0.91	1,467 mm	
Front leg length (humerus + radius + metacarpus long bones)	OR39747 (C3) vertebral body length	G. camelopardalis ontogenetic data	y = 70.2x0.598 CI = 0.332–0.8642 R2 = 0.87	1,668 mm	
Foreleg withers height	OR39747 (C3) vertebral body length	G. camelopardalis ontogenetic data	y = 7.61x0.663 CI = 0.586–0.741 R2 = 0.92	2,558 mm	
Approximate reaching height (hoof to occipital crest)	OR39747 (C3) vertebral body length	G. camelopardalis ontogenetic data	y = 7.600x0.742 CI = 0.678–0.806 R2 = 0.95	3,880 mm	

The different vertebral dimensions predict the body mass to be within a range of 228 kg–575 kg, with an average of 373 kg (Table 7, 95% confidence interval (CI) ± 168 kg). We identified which of these dimensions could predict body mass accurately across species by calculating prediction errors when applying the G. camelopardalis regression equations to both extant giraffes and okapis. Naturally, because the predictions were done using G. camelopardalis ontogenetic allometry, the G. camelopardalis prediction errors were lowest (8%–50%). Predictions for okapi body mass, however, ranged from 17% to 99%. The only variable which provided relatively low body mass prediction errors in both okapi (17%) and G. camelopardalis (25%) was the caudal vertebral body dorsoventral height. This dimension predicts a body mass of 390 kg in G. sivalensis if we consider OR39747 as a third cervical. If OR39747 was considered a fourth or fifth cervical, body mass predictions will be 274 kg (y = 0.0011x3.128) or 187 kg (y = 0.0004x3.285) respectively (Table 7).

Table 7 Functions for the prediction of body mass based on various G. sivalensis specimens.

Independent (x) variable	Model sample	Model r2	Allometric equation	Body mass prediction (kg)	Body mass PE% confidence intervals in kg (based on prediction errors when applied to G. camelopardalis data)	Body mass confidence intervals in kg (based on prediction errors when applied to O. johnstoni data	
OR39747 (C3) vertebral body length	G. camelopardalis ontogenetic data	0.91	y = 0.022∗x1.919	575	8% PE (529–612)	81% PE (109–1,041)	
OR39747 (C3) cr dv	G. camelopardalis ontogenetic data	0.77	y = 0.0023∗x3.21	400	18% PE (328–472)	87% PE (52–748)	
OR39747 (C3) cr lat	G. camelopardalis ontogenetic data	0.84	y = 0.0054∗x2.967	228	14% PE (196–260)	99% PE (2–454)	
OR39747 (C3) cd dv	G. camelopardalis ontogenetic data	0.69	y = 0.0048∗x2.847	390	25% PE (293–487)	17% PE (323–456)	
OR39747 (C3) cd lat	G. camelopardalis ontogenetic data	0.57	y = 0.0227∗x2.360	271	50% PE (136–407)	21% PE (214–328)	
Average of OR39747 vertebral dimensions (SD)				373 (135)			
OR39748 (C3) cd dv	G. camelopardalis ontogenetic data	0.69	y = 0.0048∗x2.847	394	25% PE (296–493)	17% PE (327–462)	
OR39747 (C4) cd dv	G. camelopardalis ontogenetic data	0.69	y = 0.0011∗x3.128	274			
OR39747 (C5) cd dv	G. camelopardalis ontogenetic data	0.69	y = 0.0004∗x3.285	187			
Humerus midshaft circumference (OR17136)	G. camelopardalis ontogenetic data	0.98	y = 8.96∗10−4x2.55	809	5% PE (767–851)	5% PE (766–852)	
Humerus midshaft circumference (OR39749)	G. camelopardalis ontogenetic data			772	5% PE (732–812)	5% PE (731–813)	
Average of humeral circumferences (SD)				791 (26)			
Humerus midshaft craniocaudal diameter (OR17136)	G. camelopardalis ontogenetic data	0.98	y = 3.59∗10−2x2.32	834	11% PE (743–925)	13% PE (723–945)	
Humerus midshaft craniocaudal diameter (OR39749)	G. camelopardalis ontogenetic data		y = 3.59∗10−2x2.32	602	11% PE (537–667)	13% PE (522–682)	
Humerus midshaft transverse diameter (OR17136)	G. camelopardalis ontogenetic data	0.96	y = 2.00∗10−2x2.53	561	24% PE (429–693)	22% PE (438–684)	
Humerus midshaft transverse diameter (OR39749)	G. camelopardalis ontogenetic data		y = 2.00∗10−2x2.53	813	24% PE (622–1,004)	22% PE (635–991)	
Average humeral craniocaudal and transverse (SD)				703 (141)			
All humeral ontogenetic average (SD)				732 (119)			
Radius midshaft circumference (OR17130)	G. camelopardalis ontogenetic data	0.99	y = 1.65∗10−4x2.93	1,179	10% PE (1,064–1,294)	31% PE (726–1,390)	
Radius midshaft craniocaudal diameter (OR17130)	G. camelopardalis ontogenetic data	0.98	y = 2.89∗10−3x3.19	847	12% PE (746–948)	62% PE (416–1,780)	
Radius midshaft transverse diameter (OR17130)	G. camelopardalis ontogenetic data	0.99	y = 1.18∗10−2x2.67	1,047	9% PE (948–1,146)	19% PE (943–1,387)	
Radius ontogenetic average (SD)				1,024 (167)			
Metacarpal midshaft circumference (OR39750)	G. camelopardalis ontogenetic data	0.96	y = 4.70∗10−5x3.24	1,058	11% PE (942–1,174)	31% PE (726–1,390)	
Metacarpal midshaft craniocaudal diameter (OR39750)	G. camelopardalis ontogenetic data	0.97	y = 1.59∗10−3x3.40	1,098	21% PE (867–1,329)	62% PE (416–1,780)	
Metacarpal midshaft transverse diameter (OR39750)	G. camelopardalis ontogenetic data	0.98	y = 6.71∗10−3x2.95	1,165	20% PE (932–1,398)	19% PE (943–1,387)	
Average metacarpus				1,107 (54)			
Humerus midshaft craniocaudal diameter (OR17136)	Artiodactyl interspecific allometry (Scott, 1990)	0.94	y = 7.63x2.455	1,106	18% PE (906–1,305)	24% PE (844–1,368)	
Humerus midshaft craniocaudal diameter (OR39749)	Artiodactyl interspecific allometry (Scott, 1990)			793	18% PE (650–936)	24% PE (605–981)	
Humerus midshaft transverse diameter (OR17136)	Artiodactyl static interspecific (Scott, 1990)	0.95	y = 12.4x2.46	900	26% PE (662–1,138)	52% PE (428–1,372)	
Humerus midshaft transverse diameter (OR39749)	Artiodactyl interspecific allometry (Scott, 1990)			1,268	26% PE (822–1,518)	52% (603–1,933)	
Humerus midshaft circumference (OR17136)	Various mammalian taxa (Roth, 1990)	0.99	y = 9.45∗10−4x2.61	1,170	30% PE (822–1,518)	29% PE (831–1,509)	
Humerus midshaft circumference (OR39749)	Various mammalian taxa (Roth, 1990)			1,115	30% PE (784–1,446)	29% PE (792–1,438)	
Humerus midshaft circumference (OR17136)	Various mammalian taxa (Anderson, Hall-Martin & Russell, 1985)	0.99	0.0009x2.6392	1,304	37% PE (819–1,789)	35% PE (842–1,766)	
Humerus midshaft circumference (OR39749)	Various mammalian taxa (Anderson, Hall-Martin & Russell, 1985)			1,241	37% PE (780–1,702)	35% PE (801–1,681)	
Humerus midshaft circumference (OR17136)	Ungulates (Campione & Evans, 2012)	0.95	y = 1.469x2.5273	1,167	29% PE (831–1,503)	31% PE (800–1,534)	
Humerus midshaft circumference (OR39749)	Ungulates (Campione & Evans, 2012)			1,113	29% PE (792–1,433)	31% PE (763–1,463)	
All humeral interspecific average (SD)				1,112 (180)			
Radius midshaft craniocaudal diameter (OR 17130)	Artiodactyl static allometry (Scott, 1990)	0.93	y = 29.2x2.51	1,891	50% PE (946–2,837)	54% PE (870–2,911)	
Radius midshaft transverse diameter (OR 17130)	Artiodactyl static allometry (Scott, 1990)	0.91	y = 8.19x2.555	1,238	11% PE (1,102–1,374)	43% PE (711–1,765)	
Radial interspecific average (SD)				1,565 (462)			
Notes.

SD Standard deviation

PE Prediction error

Predictions based on long bone dimensions

All of the G. sivalensis long bone specimens available at the Natural History Museum were incomplete proximally and/or distally. It was clear, nevertheless, that the bones had a similar slender appearance of extant giraffes and were elongated. Humeral specimen OR39749 was almost complete except for the proximal metaphysis, which has clearly broken off at the physeal line of a subadult animal. Regarding the radius/ulna specimen, the bones’ fusion at the midshaft was not complete as in modern giraffes, where the two bones are indistinguishable at midshaft in adults. The metacarpus specimen included in the study had the same caudal ‘columns’ or caudal groove as those evident in the extant giraffe (Solounias, 1999; Van Schalkwyk, 2004; Van Schalkwyk, Skinner & Mitchell, 2004) as well as in those of the okapi (own observation).

As no bones were complete lengthwise, bone length could not be used as a predictor for body mass which, in any case, has been shown to be a poor estimator of body mass in other taxa (Scott, 1990). Based on circumferences of the humeri (OR39749 and OR17136) and using G. camelopardalis ontogenetic data these specimens may have belonged to animals with body weight in the range of 770 kg–810 kg. An extant giraffe of this body mass would have a humerus length of about 477 mm–484 mm (y = 63.2(Bodymass)0.304), which is just slightly longer than the 453 mm measured on OR39749 that lacked a distal metaphysis. The predictors based on radial and metacarpal cross sectional dimensions offered much higher body mass estimates, with averages of 1,024 kg and 1,107 kg respectively. In addition to employing ontogenetic data to generate allometric equations, we also referred to previously published interspecific studies (Anderson, Hall-Martin & Russell, 1985; Roth, 1990; Scott, 1990; Campione & Evans, 2012). Interspecific equations tended to predict heavier body masses than ontogenetic equations, especially so in the distal long bone samples.

Predictions based on dental dimensions

Four molars and two premolars were used for size predictions (Table 4), using equations developed by Damuth (1990). Body mass predictions based on tooth length (average = 577 kg, standard deviation = 155 kg) tended to be smaller than the predictions based on tooth width (average = 881 kg, standard deviation = 188 kg, t(27) = 4.83, p < 0.01). Predictions from molar length dimensions also tended to be lower than those from premolar lengths (average from molar lengths = 473 kg, average from premolar lengths = 682 kg, t(9) = − 3.12, p < 0.05).

Discussion

Vertebral identity of OR39747

The anatomical identity of OR39747 was disputed by Lydekker (1885). He showed that Falconer was in a habit of not counting the atlas and axis as cervical vertebrae—which often meant that the start of the numbering of vertebra commenced at the third or occasionally the second postcranial vertebra. Mammalian C3–C5 forms a repetitive series and often does not have the distinguishing characteristics present in the other cervical vertebrae (Solounias, 1999). It is therefore indeed challenging to assign OR39747 to a specific vertebra. However, if we assume approximate similarity in shape between G. sivalensis and G. camelopardalis vertebrae, there are clues in the extent to which the cranial articular processes (Proc. articularis cranialis) extend beyond the body or centrum of the vertebra (Corpus vertebrae). In the G. camelopardalis C3, this process extends well beyond the cranial extremity of the vertebral body, but ends before or approximately at the same dorsoventral plane as the vertebral body in C4 and C5. Judging then by the extent of the articular processes of OR39747, it is a third, fourth or fifth cervical in decreasing order of likelihood. Falconer was therefore correct in assigning this vertebra as a third cervical, albeit fortuitously so.

Ontogenetic and interspecific scaling models

It is unusual although not unique to use ontogenetic allometry to predict an extinct animal’s size. For instance, Roth (1990) proposed that smaller animals of a species with distinctive morphologies (be they juvenile or adult) may still be better analogues than other taxa, at least in some aspects. We believe that this view is warranted in the current study as no extant species has such an extreme shape as G. camelopardalis. Predicting fossil masses from interspecific equations are further complicated by the decision of which taxa to include in regressions. For example, it is not clear whether predictions generated from interspecific allometric data are more accurate when based on closely related taxa with similar locomotor habits (Runestad, 1994; Janis, Theodor & Boisvert, 2002) or when using a wider sampling base (De Esteban-Trivigno, Mendoza & De Renzi, 2008). Other factors that may influence precision of body mass predictions in interspecific studies are body mass estimations (instead of body mass measurements), small intrataxa sample sizes, and overrepresentation of animals of one sex or of exaggerated proportions. To overcome these problems, we investigated which ontogenetic scaling parameters, if any, might be suitable and robust enough for predictions amongst extant Giraffidae. It is possible that giraffe ontogenetic equations are also acceptable for comparison not just amongst the Giraffidae but amongst, for example, extant camelids with similar gaits. However, okapis were considered as an adequate reference in this case as they are closest to giraffes phylogenetically and because the ontogenetic scaling of their long bones scales differently to other cetartiodactyla (Kilbourne & Makovicky, 2012).

Ontogenetic scaling and interspecific scaling exponents are generally not interchangeable (Gould, 1966; Pélabon et al., 2013); in this case it is dependent on the assumption that G. sivalensis had a similar body plan as juvenile extant giraffes. We thus found it appropriate, where possible, to test both ontogenetic and interspecific curves to infer proportions of G. sivalensis, but realised that neither of these methods may be appropriate for each and every dimension measured.

Neck length and reaching height

Badlangana, Adams & Manger (2009) presented interspecific predictions for vertebral neck length based on vertebral body length. Using their data (presented in Table 1 of Badlangana, Adams & Manger, 2009), we could estimate G. sivalensis C2–C7 vertebral neck length as 1,150 mm (Table 6), slightly shorter (45 mm or 4%) than vertebral neck length calculated from our ontogenetic data. There are therefore reasonable grounds to believe that our estimated neck length based on ontogenetic data is valid, or at least close to interspecific curves. Further support for this rationale can be seen where the G. camelopardalis ontogenetic curve gives appropriate predictions for vertebral neck length in both the extant giraffe and okapi (Fig. 5). Extant adult giraffes have an average external neck length of about 2,013 mm in males (1,000 kg and above) and 1,832 mm in females (800 kg and above) (Mitchell, Van Sittert & Skinner, 2009). Assuming the same body plan for G. sivalensis as for G. camelopardalis, then G. sivalensis had around 350 mm (20%) to 550 mm (27%) shorter necks than modern giraffes, depending on whether OR39747 came from a female or male animal. This is a slightly longer neck length than Falconer & Cautley’s (1843) estimated neck length for G. sivalensis, which is approximately a third shorter than extant giraffes.

Figure 5 The relationship between neck length and C3 vertebral length throughout ontogeny in giraffes and okapis.

A regression line is based on the giraffe ontogenetic series and is extrapolated to the okapi range. The use of a regression line for ontogenetic and phylogenetic allometry seems to be appropriate in this case, supporting the use of a giraffe ontogenetic regression line to predict a neck length value for G. sivalensis.

Body mass

The body mass predictions for G. sivalensis are wide (Fig. 6). Possible reasons for the large range of predictions are that certain fossils were erroneously attributed to G. sivalensis and/or that certain specimens and allometric equations are inadequate for body mass predictions. Before decisions could be made regarding the validity of attributing a fossil to G. sivalensis, we ascertained the equations that were robust enough to predict body mass accurately across species.

Figure 6 Body mass predictions for G. sivalensis based on various fossil specimens.

The labels are divided into predictions from vertebral dimensions (diamond shapes), humeral dimensions (squares), radial dimensions (circles) and metacarpal dimensions (crosses). The humeral and radial dimensions are further subdivided into those originating from ontogenetic allometric equations (red and purple, respectively) and those from interspecific equations (green and orange, respectively). Note that the interspecific predictions generally provide heavier estimates of body mass than predictions based on ontogenetic data. Furthermore, the distal bones tend to predict higher values than the proximal (humerus) bone predictions. Vertebral predictions give the lightest body mass estimates. Abbreviations: Vert, Vertebral body; H, Humerus; R, Radius; Mc, Metacarpus; Cr, Cranial; Cd, Caudal; CrTr, Cranial Transverse Dimension; CrDv, Cranial Dorsoventral Diameter; CdTr, Caudal Transverse Diameter; Cddv, Caudal Dorsoventral Diameter; Crcd, Craniocaudal Midshaft Diameter; Tr, Transverse Midshaft Diameter; Circ, Midshaft Circumference; ont, ontogenetic sample; inters, interspecific sample; Sc, (Scott, 1990); Ro, (Roth, 1990); An, (Anderson, Hall-Martin & Russell, 1985).

Vertebra OR39747 body mass estimates

It is unconventional to use vertebrae as proxies for body mass, although due to the lack of other samples it has been done before (see for instance Taylor, 2007; Taylor & Naish, 2007). As OR39747 is the holotype, it necessitates that body mass estimates are made from it if other Giraffa spp. specimens are to be attributed to it. Although vertebral body length has higher R2 values than cross sectional vertebral properties (Table 7, Van Sittert, Skinner & Mitchell, 2010), cross sectional properties are still preferable predictors of body mass in this case. The first reason is that R2 value is inferior to percent prediction error (% PE) and percent standard error of the estimate when assessing reliability of body mass predictions through regressions (Smith, 1984). Secondly, vertebral cross sectional properties are subjected to the stresses and strains within the neck (Slijper, 1946) and therefore are a much better indicator of head and neck mass and by implication body mass. Conversely, vertebral body length is influenced by factors other than body mass such as the number of vertebrae in an anatomical area (compare birds and mammals’ cervical region) or the lifestyle of the animal. We found caudal vertebral height (dorsoventral diameter) to have the lowest % PE (25% and 17%) when predicting body mass in both extant giraffes and okapis respectively (Table 7 and Fig. 7), and therefore considered this dimension to be most robust for body mass predictions across giraffids. There are no other published interspecific regression equations using vertebral dimensions for the prediction of body mass in ungulates of which we are aware. The caudal vertebral height predicts a body mass of 390 kg in G. sivalensis. Interestingly, the average body mass prediction from the remaining vertebral regression equations (C3 vertebral body length, cranial height, cranial width and caudal width, Table 7) is fairly similar—368 kg. The only body mass prediction to fall outside the 95% confidence interval based on all vertebral dimensions including vertebral height (373 kg ± 119 kg) is vertebral body length, predicting a mass of 575 kg.

Figure 7 The body mass prediction errors (absolute values) associated with various dimensions in Okapia johnstoni and Giraffa camelopardalis.

Of the available regressions and variables measured, it would appear that humeral circumference and craniocaudal diameter (using G. camelopardalis ontogenetic regression) is best suited for body mass predictions, both in giraffes and okapis, and therefore also likely to be useful for body mass predictions in G. sivalensis. Vertebral caudal dorsoventral diameter represents an acceptable variable should estimates only be based on the holotype, with prediction errors of 17% and 25% in giraffes and okapis respectively. Different shapes indicate different bones used for body mass predictions. Note that for clarity of the graph, the maximum indicated prediction error is 100%. Abbreviations: Oj, Okapia johnstoni; Gc, Giraffa camelopardalis; P.E, Prediction Error; other abbreviations as listed for Fig. 2.

Nevertheless, the body mass prediction from caudal vertebral height could be either an over or underestimate. Considering it as an overestimate would mean that this animal had a relatively heavy neck and head complex but a slender or lightweight body. This is unlikely as a larger head and neck complex is unsupportable unless accompanied by a larger total body size (Taylor & Wedel, 2013). Conversely an underestimate would mean a slender neck and head complex but a relatively stocky body. This is a more plausible scenario and if indeed it is the case, it might explain the discrepancy between vertebral and dental body mass predictions when compared to those of limb bones.

Limb bone body mass estimates

Interspecific long bone cross sectional properties, although probably more closely related to body mass than any other variable, have nevertheless been found to be poor predictors of body mass in giraffes and in some cases, okapis (McMahon, 1975; Anderson, Hall-Martin & Russell, 1985; Scott, 1990; Janis, Theodor & Boisvert, 2002), although it should be noted that a recent interspecific study has shown giraffes to be more amenable to interspecific equation predictions (Campione & Evans, 2012). Similarly, we found higher prediction errors with interspecific equations compared to G. camelopardalis ontogenetic curves, with a 5% prediction error based on humeral ontogenetic data (Fig. 7). Errors were inflated when using more distal bones. Therefore, the most appropriate long bone variable useful for G. sivalensis body mass determination is very likely humeral cross sectional properties, using our ontogenetic G. camelopardalis sample.

The average body mass estimated from humeral ontogenetic analysis is 732 kg. Interestingly, this body mass is about 150 kg more than would be indicated by a G. camelopardalis of similar neck length, and 342 kg more than the mass predicted from OR39747 cross sectional properties. This could mean that either the humeral fossil specimens were incorrectly assigned to G. sivalensis, that G. sivalensis had a relatively stockier body and thinner neck than G. camelopardalis or that the holotype vertebra came from a female and the humeral specimens from large males.

Unfortunately, none of the other long bone dimensions seem to be reliable predictors of body mass across extant giraffids. The best non-humerus candidate using interspecific scaling seems to be the radius transverse diameter with a 43% and 11% prediction error in okapis and giraffes respectively. This dimension predicts that the specimen belonged to an animal of approximately 1,238 kg, which suggests this animal might have been heavier than G. sivalensis. There are no interspecific equations for metacarpi that we could find, and therefore we could only rely on ontogenetic equations. Yet, similar to the radial prediction, the metacarpal transverse diameter predicts a body mass of 1,165 with around 20% prediction error. The inflated prediction errors could be because humeri and femora are generally more suitable for body mass predictions than more distal bones, especially in giraffes (McMahon, 1975). It is also possible that the fossil long bones were incorrectly assigned to G. sivalensis and perhaps belonged to another similar species existing at the same time and location.

Dental body mass estimates

There have been numerous dental specimens ascribed to G. sivalensis (Table S1). Unfortunately, not all of these specimens are from the same locality and are probably from different stratigraphic zones. Subsequently, there appeared to be uncertainty regarding the correct species allocation of these fossils (see especially Lydekker, 1876). A discussion on the morphology and correct species classification of teeth specimens assigned to G. sivalensis were not considered as part of this study, and we therefore used only those teeth mentioned by Falconer & Cautley (1843). These specimens were originally assigned to the species G. affinis—a classification later abandoned by Falconer himself and also disputed by Lydekker (1883), who re-assigned the fossils to G. sivalensis. As the specimens originated from the same area and strata as the holotype OR39747, which is the Pliocene of the Siwaliks (Lydekker, 1885), we believe it reasonable to consider them as truly G. sivalensis teeth until further evidence emerges.

Molar length measurements are more reliable indicators of body mass than molar width or area (Damuth, 1990; Fortelius, 1990; Janis, 1990). Furthermore, Janis (1990) found that premolar row length are poorer correlates than molar row length. Molar lengths predict an animal within the range of 288 kg–673 kg, which is similar to OR39747’s caudal vertebral height body mass prediction of 390 kg.

Combined size estimates

Lydekker’s (1885) suggestion that OR39747 belonged to a small individual could have meant that the animal was still immature, that the animal was a relatively small individual of the species or that the species itself was small within the genus. It is unlikely that Lydekker meant an immature animal as the fusion of the epiphyses to the body of the vertebra is complete and clear definitions of bony ridges and muscular depressions indicate a mature animal (Falconer & Cautley, 1843). Lydekker might have based his idea of a small individual on two larger vertebrae assigned to G. sivalensis—a proximal part of a ‘third’ and distal part of a ‘fourth’ cervical, OR39746 and OR39748 respectively (Lydekker, 1885, Table S1). Unfortunately, these vertebrae were not locatable within the Siwalik collection at the time of this study (P Brewer, pers. comm., 2013, Curator of fossil mammals, Natural History Museum), and we subsequently could not measure them. Nevertheless, Falconer (1845) reported OR39748 to be 2.1 inch (53.3 mm) in height and width at the caudal extremity, which is only 0.2 mm greater and 0.1 mm less than our respective measurements of OR39747 (Table 2). Based on ontogenetic allometry for caudal vertebral body height, OR39748 came from an animal weighing 394 kg or 277 kg, depending on whether it was a C3 or C4 vertebra respectively (Table 7). The animal from which the holotype vertebrae originated was therefore also not relatively small compared to the size estimated from specimen OR39748. It is possible though, especially considering body mass estimates from the humerus, that there might have been sexual size dimorphism present in G. sivalensis. If that is indeed the case, OR39747 and OR39748 would have been females about half the size of fully grown male animals, a possibility also supported by the fossil teeth considered in body mass estimates.

Conclusion

Our considered opinion is that the G. sivalensis, from which the holotype cervical vertebra originated weighed approximately 400 kg, had a neck length of about 1.47 m and a reaching height of 3.9 m. There is a possibility that it displayed sexual dimorphism, in which case male animals would have been a little less than twice the size of females and both would have had a similar morphology. If sexual dimorphism was not present and all bones were correctly attributed to this species, then the animal had a slender neck with a relatively stocky body, a shape that is not unrealistic to imagine.

Supplemental Information

Data S1 Raw data pertaining to measurements on G. sivalensis specimens as well as Okapi johnstoni vertebrae

Click here for additional data file.

Table S1 Fossil specimens that have been assigned as belonging to Giraffa sivalensis

Click here for additional data file.

We thank Virgini Volpato and Katrin Krohman for assistance at the Senckenberg Museum, Frankfurt; Emma Bernard for assistance at the Natural History Museum, London; Tarik Afoukati from the Muséum national d’Histoire naturelle, Paris and the Ditsong Museum of Natural History, Pretoria, South Africa. We are grateful to Ms Marie Watson at the Centre for Veterinary Wildlife Studies for her general administrative assistance in the planning of travel to relevant institutions. Thank you to the reviewers of this manuscript for valid and valuable input. The late Prof John Skinner’s support was instrumental in the completion of this study.

Additional Information and Declarations

Competing Interests

Author Contributions

The authors declare there are no competing interests.

Sybrand J. van Sittert conceived and designed the experiments, performed the experiments, analyzed the data, contributed reagents/materials/analysis tools, wrote the paper, prepared figures and/or tables, reviewed drafts of the paper.

Graham Mitchell conceived and designed the experiments, contributed reagents/materials/analysis tools, wrote the paper, reviewed drafts of the paper.

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
