# Peer review of "On reconstructing Giraffa sivalensis, an extinct giraffid from the Siwalik Hills, India"

_PeerJ, doi:10.7717/peerj.1135_

## Round 0.1 · original submission · Major Revisions

Hello, we obtained three speedy and constructive reviews for this manuscript and there's a lot to consider in revisions but it all seems quite helpful feedback to me. Be sure to address all points individually, in detail, in a Response document when you submit your revised MS, as reviewers often focus on that document when they re-review a MS, and this MS will be sent back for review upon resubmission of the revision. But all 3 reviewers agree there is something here that deserves eventual publication. Well done!

·

Basic reporting

The paper meets most of the basic criteria - the questions are straightforward: what is the estimated body mass of the fossil giraffe G. sivalensis, and does the material referred to that species represent a reasonable size range or have some elements been mistakenly referred to G. sivalensis that clearly represent a second giraffe with dimensions falling outside that range?

Overall the analyses and data reported are appropriate to answer the question and detail the history of these specimens nicely, it just needs a little fleshing out.

Experimental design

In my opinion, there needs to be a more complete discussion of why ontogenetic regression data are appropriate to predict mass in okapi (different genus). I can see the argument that giraffe is tough to predict using regressions from other ruminants, but am not so sure they have made a solid argument for the use of ontogenetic regression for giraffe in okapi.
While okapi and giraffe are sister taxa, there are major differences in size and structure. It is usually assumed that okapi are more similar to other ruminants than giraffe - certainly true in other postcranial dimensions (see Janis et al 2002, which I will attach). Is it the case that neck length in okapi is the same scaling as giraffe, where other dimensions follow the relationships for ruminants? What about using camelid data for postcranial estimation of Giraffa, given their locomotor similarities?

Validity of the findings

The results section comparing various postcranial mass predictors suffers from not including dental predictions into the comparison - since this is the main method used in fossil mass prediction for mammals it is important to see how it compares with the postcranial measures used here. It is clear from the supplemental Table 1 that some dental material does exist, and it would be very valuable to see what the results of mass estimation from them are and how they compare with the postcranial (both ontogenetic and interspecific) predictions made in this paper.

Additional comments

There are numerous grammatical and style inconsistencies throughout the MS as noted in the attached MS file. Please edit for consistency in usage and clarity.
I was torn between calling the above major vs. minor revisions - I'd call them moderate.

Reviewer 2 ·

Basic reporting

article is fine - no comments

Experimental design

design is fine I see no flaws in the body weight-size prediction

Validity of the findings

the findings seem valid

Additional comments

Review for G. sivalensis
This study is about body size prediction of an extinct giraffid species. This should be stated in the title. Title: Reconstructing the size of… This is all it does. As such I find the study personally as not of a significant importance. Others may find it significant. The size is so similar to that of a small giraffe that it really makes no difference what the actual weight prediction is. What is bothersome is that they assume these bones found so long ago come from the same individual and the same horizon. Their findings show an inconsistency with a smaller cervical than limbs (last page). They assume they belong to one individual. They also assume that the specimens all belong to the same species. It is known that during the Mio-Pliocene there were usually three or more giraffid species per locality. If these are indeed all from one taxon, why does it have to have the proportions of a modern giraffe? Look at Litocranius and Gazella granti. If you found partial limbs and a cervical of each you would be very confused about proportions. And these two species coexist. They never mention the dentition of sivalensis in the same drawer. The dentition is actually a good predictor of size and as such one would be interested in their assessment of the teeth.
The second problem of this study is how much they cover and discuss of terms very unfamiliar to me: Ontogenetic data, ontogenetic sample, for allometric equation, ontogenetic equations and interspecific curves. I assume other readers will also wander about these terms. Why so much ontogeny when considering a fossil adult vertebra? I have not seen other studies like this? It is not a juvenile vertebra and so be it. Ontogeny for fossils is of remote usefulness for this particular case; where we do not even know what we have? And is it Giraffa? This is another problem.
I suppose the size predictions in principle are fine although I do not understand the application of ontogeny here. People who have been working mostly with recent animals have a problem when they leap into the paleontological research and vice versa.
Miscellaneous comments:
196 – 205 Ontogenetic data ontogenetic sample for allometric equation and ontogenetic equations
241 – 254 allometric problem – what problem explain
261-264 interpsecifc curves – what is this?
312 conflicting data on size
Capitalize Siwalik Hills
Lines: 35-36 – 50 – 51 confusing re-write and explain better
48 G. attica and G. vetusta are a very arcane names – remove from your list – these are Bohlinia attica now and hence not Giraffa
48-49 It seems like you are conducting a quick systematic revision. This is a rather involved task. You cannot say all these names can be or may be sivalensis. I would add a clarification that you are speculating here.
50 there is no controversy as to the prevalence of giraffids in the Siwaliks – be precise
65 what is 0.6 million year? Write a number
75 or distinct genera – very unclear – what do you mean?
95 The Natural History Museum
149-157 – very minor need to mention these observations
151 – British museum is that of archaeology in London – and Museum should be: The Natural History Museum
256 giraffines this is vague – do you mean Giraffinae?
Sir Cautley – when a person has died we remove titles – unless you are writing a historical type of text
218 – proecessus arrticularis cranialis is the only anatomical term you use. Why so fancy with this there are dozens of other characters on vertebrae that you ignore.
231- vertebrae (plural)
Capitalize: (island) - Perim Island

Reviewer 3 ·

Basic reporting

All estimates of G.sivalensis mass or length should be accompanied by confidence intervals, especially in the abstract. The estimates should also have a reference attached, with the referenced allometric equation. These are interesting findings; stating the equation used makes it much easier for the reader to follow the thought process.

In at least two places (157 and 252), assumptions are stated, which affect the validity of applying allometric models to G.sivalensis. It would be clearer if any assumptions are explicitly stated in their own section, either in the methods or discussion. This would also allow inferences to read in a more straightforward manner (instead of having to repeatedly prefix a limitation in the same sentence as a conclusion).

Experimental design

The interspecific scaling studies referenced here are very useful papers, but they carry their limitations, which have been previously documented by others (see the discussion by Campione and Evans 2012). I recommend including and using the scaling equations from Campione’s study, as they addressed and attempted to correct for much of these limitations.

Validity of the findings

The historical account that the holotype is from ‘a very small individual’ is a significant point. Given that the mass estimate from OR39747 is one of the cornerstones of this paper, any controversy surrounding the specimen should be declared early on. Although this issue is at present addressed in the discussion, the reader should know about this apparent controversy before reading the results section. I also think that it’s worth revisiting this towards the end of the discussion, as the results suggest that the holotype does actually come from an animal which is of smaller body mass than the owner of the humeral specimens.

Additional comments

This is a useful insight into a historically significant extinct giraffid.

Specific comments:
31. Ambiguous taxonomy. Uninitiated readers might get confused and think that G.sivalensis was the first extinct giraffid to be described. Consider replacing ‘giraffe’ with ‘Giraffa’. Using the phrase ‘longest established’ is also ambiguous, does this mean it was the first to be described, or the earliest known genus?

84. Lydekker described the holotype as being from ‘a very small individual’. This statement is examined in the discussion section, but I think this should be brought up sooner; either here or in the introduction.

91. These dimensions should be demonstrated onto an annotated figure (eg. Figure 2)

93. OR39749 is another bone from a ‘small individual’. Again, this is discussed later, but for the sake of clarity, it will be useful to mention this at an earlier point. (summarize in table form?)

110. These sources of allometric equations are very useful, but carry their limitations. For example, none of the interspecific relationships are adjusted to account for phylogenetic bias. I recommend including mass estimates using equations from Campione 2012. This is an interspecific scaling study, looking at a diverse sample of tetrapods. They provide a universal scaling relationship for body mass vs humeral/femoral circumference, with an investigation into the effect of phylogeny. Usefully, they also include scaling relationships for different orders (eg. ungulate specific allometry), and then sub-categorize these by taxa of different size. Giraffes and okapi are included in the dataset. They summarize and address criticisms of Anderson et al’s dataset (Anderson, Hall-Martin et al. 1985).

116. Log transformation is the standard in scaling studies, but recently the method has come into question, with concerns about bias. This is an ongoing discussion, eg. (Packard 2013, Mascaro, Litton et al. 2014), but warrants mention.

124. Are okapi placed in Giraffinae? Solounias (Solounias 2007) places them in the ‘Okapiinae’ group. An assignment of okapi into Giraffinae should be referenced.

133. What is the reason for choosing 20% as a cut off?

136. All estimated parameters (eg. neck length, body mass) should be accompanied by confidence intervals, or some other indication of error.

138-146. It’s not clear how this section relates to the rest of the study. Does the geology type affect the results of this study?

152. (Table 3). A reference or allometric equation should accompany the predicted dimensions.

154. The ontogenetic scaling relationship should be referenced.

157. The use of the phrase ‘we could estimate..’ suggests a higher degree of uncertainty than other sections. The text then explains this uncertainty by stating an assumption, which is that G.sivalensis dimensions can be modelled suitably from G.camelopardalis ontogeny. This assumption is built into the whole methodology (ie. predicting extinct taxa neck length and body mass using giraffe ontogeny). This section of the results relies on the same assumptions as the rest of the paper, so the language used here can be more confident, as long as the limitations of the predictions are explicitly stated.

160. Refer to Table 5.

176. As far as I can see, the ‘columns’ aren’t mentioned in van Schalkwyk, Skinner et al. 2004, but instead in van Schalkwyk’s university dissertation from 2004. This morphology (the posterior groove of the metapodial bones?) is also described in Solounias 2007.

180. The text here doesn’t specify that OR17130 refers to a radius/ulna.

182. A reference for the scaling equation should accompany this length estimation.

192. As before, this dispute should be stated earlier, before the reader has seen all the results derived from the holotype specimen.

213 - 218. Clarify that it was the individual animal possessing OR39748, which would have weighed 394 or 277kg. How was the mass prediction calculated? The scaling equation used here should be referenced, and the metric used (vertebral height?) should be specified.

The data here isn’t able to support this conclusion. The data doesn’t dispute the claim that the holotype came from a ‘small individual’, instead it suggests that the holotype wasn’t much smaller than comparable vertebral specimens.

This section in general is hard to follow. It would help if the estimates were referenced, and if the body mass calculations for OR39748 were better explained (consider including them in a table).

The conclusion here is also based on vertebral width. Length may be a better indicator of body mass (given that R2 values are higher than with vertebral width,(Van Sittert, Skinner et al. 2010). Of course this isn’t possible, as OR39746 and OR39748 are incomplete (and lost), but this is worthy of a mention.

220. The method of using ontogenetic scaling to predict body mass isn’t unique (eg. Roth 1990), but studying Giraffa represents a good opportunity to use this method, as giraffe body shapes are so specialized compared to other ungulates.
Should specify that the study is using allometry to predict an *extinct* adult animal’s size.

226. The methods used here should also assess the validity of recent interspecific studies, such as Campione 2012.

254. Should read ‘this is close to…’.

273. Vertebral cross sectional properties are better than…? It would be clearer to clarify what dimension they’re preferable to. But, this may not be the same in giraffes.

280. Refer to table 5.

284-296. There are a lot of ‘if’ statements, overestimates, underestimates, and slender/stocky anatomy in the same paragraph. This section could be made clearer by separating out the arguments, or even omitting some of them. The overall message of this section is a bit lost in the detail.

286. The text uses the word ‘either’, which reads strangely without later adding an ‘or’ clause.

287. Remove the ~s from the end of ‘overestimates’.

309. This implies that perhaps the holotype does in fact come from a very small individual.

432 Misspelling of ‘Stuart’ (written Suart), and misspelling of ‘Field’


Anderson, J. F., A. Hall-Martin and D. A. Russell (1985). "Long-bone circumference and weight in mammals, birds and dinosaurs." Journal of Zoology 207(1): 53-61.
Campione, N. E. and D. C. Evans (2012). "A universal scaling relationship between body mass and proximal limb bone dimensions in quadrupedal terrestrial tetrapods." BMC Biol 10: 60.
Mascaro, J., C. M. Litton, R. F. Hughes, A. Uowolo and S. A. Schnitzer (2014). "Is logarithmic transformation necessary in allometry? Ten, one-hundred, one-thousand-times yes." Biological Journal of the Linnean Society 111(1): 230-233.
Packard, G. C. (2013). "Is logarithmic transformation necessary in allometry?" Biological Journal of the Linnean Society 109(2): 476-486.
Roth, V. L. (1990). "Insular dwarf elephants: a case study in body mass estimation and ecological inference." Body size in mammalian paleobiology: estimation and biological implications. Cambridge University Press, New York: 151-179.
Sellers, W. I., J. Hepworth-Bell, P. L. Falkingham, K. T. Bates, C. A. Brassey, V. M. Egerton and P. L. Manning (2012). Minimum convex hull mass estimations of complete mounted skeletons.
Solounias, N. (2007). "Family Giraffidae." The evolution of artiodactyls: 257.
van Schalkwyk, O. L., J. D. Skinner and G. Mitchell (2004). "A comparison of the bone density and morphology of giraffe (Giraffa camelopardalis) and buffalo (Syncerus caffer) skeletons." Journal of Zoology 264(3): 307-315.
Van Sittert, S. J., J. D. Skinner and G. Mitchell (2010). "From fetus to adult—an allometric analysis of the giraffe vertebral column." Journal of Experimental Zoology Part B: Molecular and Developmental Evolution 314(6): 469-479.

---

## Round 0.2 · Minor Revisions

Congratulations- you have satisfied all 3 reviewers except for 1 ultimate issue which is to give the paper 1 final pass to correct taxonomy and other wording and thus give the prose a fine professional sheen so that we can publish it. I do not anticipate needing further review once the revised MS is submitted, but please Track Changes so I can check the revisions quickly and then presumably accept the paper if I'm satisfied with the final changes.

·

Basic reporting

No additional comments

Experimental design

No comments

Validity of the findings

No comments

Additional comments

Overall the authors appear to have satisfied my concerns, and addressed those of the other reviewers. However, there is still sloppy taxonomic usage - please be consistent when using terms - for example Giraffidae is capitalized, but giraffids should not be, but is sometimes capitalized within the MS.

Reviewer 2 ·

Basic reporting

I find the article as acceptable now as it is

Experimental design

The statistical methods of body estimation seem fine

Validity of the findings

their body-size estimates seem valid

Additional comments

As long as you sate (and you have) that you are projecting from one species to another - your methods seem valid

Reviewer 3 ·

Basic reporting

No futher comments

Experimental design

No futher comments

Validity of the findings

No futher comments

Additional comments

Thank you for your considered approach to these revisions.

---

## Round 0.3 · accepted · Accept

Well done - congratulations on having this paper accepted!